# A Spectral–Spatial Transformer Fusion Method for Hyperspectral Video Tracking

**Ye Wang, Yuheng Liu, Mingyang Ma and Shaohui Mei *** 

School of Electronics and Information, Northwestern Polytechnical University, Xi'an 710129, China; wy2017263322@mail.nwpu.edu.cn (Y.W.); hnlyh@mail.nwpu.edu.cn (Y.L.); mamingyang@mail.nwpu.edu.cn (M.M.)
* Correspondence: meish@nwpu.edu.cn

**Abstract:** Hyperspectral videos (HSVs) can record more adequate detail clues than other videos, which is especially beneficial in cases of abundant spectral information. Although traditional methods based on correlation filters (CFs) employed to explore spectral information locally achieve promising results, their performances are limited by ignoring global information. In this paper, a joint spectral–spatial information method, named spectral–spatial transformer-based feature fusion tracker (SSTFT), is proposed for hyperspectral video tracking, which is capable of utilizing spectral–spatial features and considering global interactions. Specifically, the feature extraction module employs two parallel branches to extract multiple-level coarse-grained and fine-grained spectral–spatial features, which are fused with adaptive weights. The extracted features are further fused with the context fusion module based on a transformer with the hyperspectral self-attention (HSA) and hyperspectral cross-attention (HCA), which are designed to capture the self-context feature interaction and the cross-context feature interaction, respectively. Furthermore, an adaptive dynamic template updating strategy is used to update the template bounding box based on the prediction score. The extensive experimental results on benchmark hyperspectral video tracking datasets demonstrated that the proposed SSTFT outperforms the state-of-the-art methods in both precision and speed.

**Keywords:** transformer fusion; spectral–spatial joint; hyperspectral object tracking

## 1. Introduction

Object tracking is one of the most important research fields in computer vision [1] and has been widely developed in the past decade and successfully employed in many applications [2], such as video surveillance [3], artificial intelligence [4], intelligent traffic control [5], and autonomous driving [6]. It is a challenging task that requires constantly tracking the object in the video subsequences due to the fast motion, occlusion and interference from similar appearances, to name a few. Most of the existing methods merely obtain color intensities, texture, semantic information, and deep representation features to track the object in the limitations of imaging technology, which are not suitable for HSVs tracking. Thanks to the advent of the hyperspectral imager, object tracking has been extended to hyperspectral images, which is some of the best research because hyperspectral images have a large amount of spectral information. Many efforts have been made to improve the performance of hyperspectral video tracking in recent year; however, the performance of hyperspectral video tracking is still unsatisfactory in complex scenarios in that the spectral information of hyperspectral images is not fully utilized in the existing object tracking methods.

Spectral reflectance preserved in hyperspectral images contains a unique feature of the hyperspectral object potentially, which is not available in other types of images [7–9]. Hyperspectral images record high-dimensional spectral information [10–12], which is advantageous for the discriminative feature extraction of objects in challenging tracking extraction such as similar appearances, and scale change. Previous studies revealed that [13]

spatial–spectral information can increase inter-object separability and discriminability to handle the tracking drift. At present, traditional trackers focus on optical feature extraction, which is not suitable for hyperspectral video tracking. However, the performance of existing hyperspectral video trackers is still unsatisfactory in typical complex scenarios. The first reason is that the existing hyperspectral trackers have not fully explored the spectral–spatial information to describe HSVs [14]. Chen et al. [13] proposed a fast spectral–spatial convolution kernel feature-extraction method to extract the discriminative feature of hyperspectral images. However, the proposed method merely tends to extract efficient encoding of local spectral–spatial information rather than the global feature maps. Therefore, the performance of the proposed method is still poor in challenging scenarios on account of the lack of global interactivity of data. Meanwhile, the proposed method is most appropriate for specific public datasets, while the tracking results display poor generalization ability. The second reason is that high-dimensional spectral information is a double-edged sword, which will bring difficulties to the feature extraction and high computational costs due to the enhancement of the bands. Xiong et al. [15] proposed two spectral–spatial feature extractors, namely, local spectral–spatial histogram of multidimensional gradient (SSHMG) and spatial distribution of materials (MHT). The former method captures spectral–spatial texture information using gradient orientations, and the latter method obtains two visual feature descriptors, namely end-members and abundances, yielding material to track. Nevertheless, the most attention in the MHT is concentrated on spectral extraction, resulting in spending massive computation costs to extract the feature information, which is not suitable for real-time tracking.

In recent years, Uzkent et al. [14] produced a synthetic aerial hyperspectral dataset with the digital imaging and remote sensing image generation (DIRSIG) software. However, the dataset is generated at 1.42 fps, leading to the difficulty of tracking objects which change rapidly in a short period of time. Thanks to the development of hyperspectral imaging technology, hyperspectral video has been widely obtained in various scenarios. The existing two types of hyperspectral video datasets are collected by [13,15], named IMEC16 and IMEC25, respectively. Accordingly, hyperspectral trackers have been developed rapidly.

Among the present hyperspectral trackers, correlation filters (CF) [14,16] and discriminative correlation filters (DCF) [17,18] have achieved much success in terms of tracking. These trackers learn an object prediction model for location in video subsequences by using the correlation-minimizing object function, which integrates both foreground and background knowledge, providing effective features responding to the model. Xiong et al. [15] proposed two spectral–spatial feature extractors, namely SSHMG and MHT, which are embedded into background-aware correlation filters to track specific objects. SSHMG is designed to capture local spectral–spatial histograms of multidimensional gradients, while MHT is proposed to represent material information, and uses fractional abundances to encode the material distribution. Liu et al. [19] introduced a spectral classification branch into the anchor-free Siamese network to enhance the representation of objects in HSVs. Lei et al. [20] employed a spatial–spectral cross-correlation embedded dual-transfer network (SSDT-Net) to extract high-dimensional characteristics of HSVs. Meanwhile, a spatial–spectral cross-correlation module is designed to capture material information and spatial distribution with two branches of the Siamese network. Zhang et al. [21] proposed spectral matching reduction features and adaptive-scale 3D hog features to track the objects to confront scale variation, where adaptive-scale 3D hog features mainly consist of cube-level features at three different scales. Zhao et al. [22] proposed a feature fusion network for the synchronous extraction of the spatial and spectral features of hyperspectral data, where the color intensity feature and the modality-specific feature are mixed to assist the tracker in accurate positioning. However, the method also displays limitations in that the severe inductive bias of CF and DCF is imposed on the model, which leads to poor generalization performance. Consequently, the model is only suitable for the object in available data but it could not integrate any learned priors. At present, several deep learning-based methods have gradually presented. Liu et al. [23] proposed a dual deep Siamese network framework with a pretrained RGB tracker and

spatial–spectral cross-attention learning. Afterward, they [24] proposed unsupervised deep learning-based object tracking framework, and a new hyperspectral dataset was collected. Nonetheless, the aforementioned methods are merely verified on a public hyperspectral video dataset [15]. The further generalization ability of these models must be investigated for generating a universal hyperspectral video tracker. Although Chen et al. [25] proposed a feature descriptor using a Histogram Oriented Mosaic Gradient (HOMG) to gain spatial–spectral features directly from mosaic spectral images on two datasets, the performance of the tracker displays a preference for IMEC25. On the contrary, transformers have shown a strong global reasoning capability across multiple frames, which is a great advantage for video tracking. Specifically, the self-attention and the cross-attention mechanism can capture the global interaction of the video with considerable success [26–28]. However, the existing transformer-based methods specially designed for natural scenes merely focus on extracting spatial features, and are not suitable for hyperspectral video tracking due to the ignorance of spectral information. An effective approach to address this issue is integrating spectral information into RGB trackers to achieve multi-modal tracking. For example, Zhao et al. [29] proposed a tracker named TMTNet to efficiently transfer the information of multi-modality data composed of RGB and the hyperspectrum in the hyperspectral tracking process. Nevertheless, these methods rely on consistent, aligned RGB and hyperspectral images, which are generally unavailable in practice in real scenes.

The transformer, benefiting from excellent global interaction ability and generalization performance, has been introduced to improve the accuracy of tracking. It is worth noting that the transformer is a novel structure that has not been applied to universal hyperspectral video tracking. It is necessary to illustrate that the transformer could be utilized to extract the spectral–spatial features of hyperspectral video sequences. The transformer was first introduced by Vaswani et al. [30] to deal with sequential tasks. It is customary that the sequences-to-sequences structures are isomorphic across layers, and the success of multi-head self-attentions (MSAs) for computer vision is now indisputable [31]. The self-attentions [32] aggregate spatial tokens that can be unified into a single function:

$$\mathcal{A}_j = \sum_i Softmax(\frac{QK^T}{\sqrt{d}})_i V_{i,j} \tag{1}$$

where $Q, K$, and $V$ are defined as query, key, and value, respectively. $d$ is the dimension of the query and key, Equation (1) is a simple function that can be used to calculate the attention score of the image token. A single-head self-attention layer limits the ability to focus on one or more specific positions. Multi-head attention is a mechanism that can be used to boost the performance of the self-attention layer [33]. The multi-head self-attention process is as follows:

$$MultiHead(Q', K', V') = Concat(head_1, \dots, head_h)W^O$$
$$head_h = \mathcal{A}_h(QW_h^Q, KW_h^K, VW_h^V) \tag{2}$$

where $Q'$ ($K', V'$) is the concatenation of the query (key, value) vectors of all heads, $W^O$ is the projection matrix of the output and $W_h^Q, W_h^K$, and $W_h^V$ are the projection matrices of the query, key, and value, respectively.

Several transformer-based trackers [1,26,28] have been proposed to deal with the tracking task. The transformers are typically employed to predict discriminative features to localize the object. Cao et al. [34] designed an adaptive temporal transformer to encode temporal knowledge effectively before the temporal knowledge is decoded for accurate adjustment of the similarity map. Mayer et al. [35] proposed a transformer-based tracker, where transformers obtain global relations with a weak induction bias, allowing the prediction of more powerful target models. Bin et al. [27] proposed an encoder-decoder transformer without using any proposals or predefined anchors to estimate the corners of objects directly, where the prediction head is a simple fully convolutional network. Wang et al. [26] separated the transformer encoder and decoder into two parallel branches

designed within the Siamese-like tracking. The encoder promotes the target templates via attention-based feature reinforcement, while the decoder propagates the tracking cues from previous templates to the current frame.

Existing transformer-based trackers are mainly designed for optical (red, blue, and green) videos or multi-modal videos. In hyperspectral videos, there is abundant spectral information and context semantics among successive frames, which have been largely overlooked in transformer-based trackers. Therefore, it is significant to design a transformer-based tracker to deal with the hyperspectral video tracking task.

In this paper, a novel hyperspectral video tracker named Spectral-Spatial Transformer-based Feature Fusion Tracker (SSTFT) is proposed to address the aforementioned problems by adequately utilizing the spectral–spatial information of hyperspectral images. As the spectral–spatial information is separate in the original hyperspectral images, the proposed SSTFT adopts a shallow spectral–spatial (SSS) subbranch and a deep ResNet (DRN) subbranch to integrate multi-scale representations and promote preliminary interaction of information. On this basis, a transformer-based context fusion (TCF) module is designed to adequately fuse the template branch and the search branch features, which can effectively establish the context relationship of hyperspectral video sequences. Furthermore, an adaptive dynamic template update (ADTU) module is designed to deal with the problems of object drift in challenging conditions.

To summarize, compared with the existing hyperspectral object trackers, the main contributions of this paper are as follows.

1.  A spectral–spatial multiscale feature extraction module is proposed for the adaptive fusion of multiple-level semantics of hyperspectral videos. Two parallel branches are designed to extract the features with full integration of spectral and spatial information, and each branch is further divided into multiple levels to extract the semantic information. The fusion strategy of coarse-grained and fine-grained features can effectively improve the completeness of the representation of objects and backgrounds.
2.  A context fusion module based on the transformer is proposed to fuse the template branch and the search branch features, which can effectively establish the forward and backward frame-dependence relationship of hyperspectral video sequences with the ranking of prediction box confidence scores rather than cosine penalty. Meanwhile, the proposed method can capture the global interaction of the video with considerable success, improving the robustness performance of the hyperspectral tracker.
3.  An adaptive dynamic template update strategy is proposed to handle the drift of object regression in challenging scenarios. The approach ensures the tracker can adapt to the changing environment and further improves the robustness of performance.

The rest of the paper is structured as follows. The proposed SSTFT tracker is detailed in Section 2. The experimental results are presented in Section 3. Finally, the conclusions are drawn in Section 4.

## 2. Methodology

The proposed spectral–spatial transformer-based feature fusion tracker (SSTFT) is introduced in this section in detail. As shown in Figure 1, our model consists of three parts, a spectral–spatial multiscale feature extraction module, a transformer-based context fusion module, and an adaptive dynamic template update strategy. It is difficult to train a robust transformer network to extract deep features from hyperspectral videos, which is mainly due to the shortage of public hyperspectral videos. Therefore, the whole structure of the proposed method utilizes the pre-trained model [1] to prevent overfitting and poor generalization ability [36].

The spectral–spatial multiscale feature extraction module is proposed to extract the multiple-level features with a shallow spectral–spatial (SSS) subbranch and a deep ResNet (DRN) subbranch, both of which are designed for multiple spectral bands. The transformer-based context fusion (TCF) module is proposed to fuse the spectral–spatial features from the template branch and search branch with the transformer integrated with self-attention

and cross-attention. Self-attention is utilized to realize the feature interaction of the systems own context within each branch. However, cross-attention is utilized to implement the feature interaction between the template branch and the search branch. The adaptive dynamic template update (ADTU) strategy is proposed to update the template with more reliable response scores.

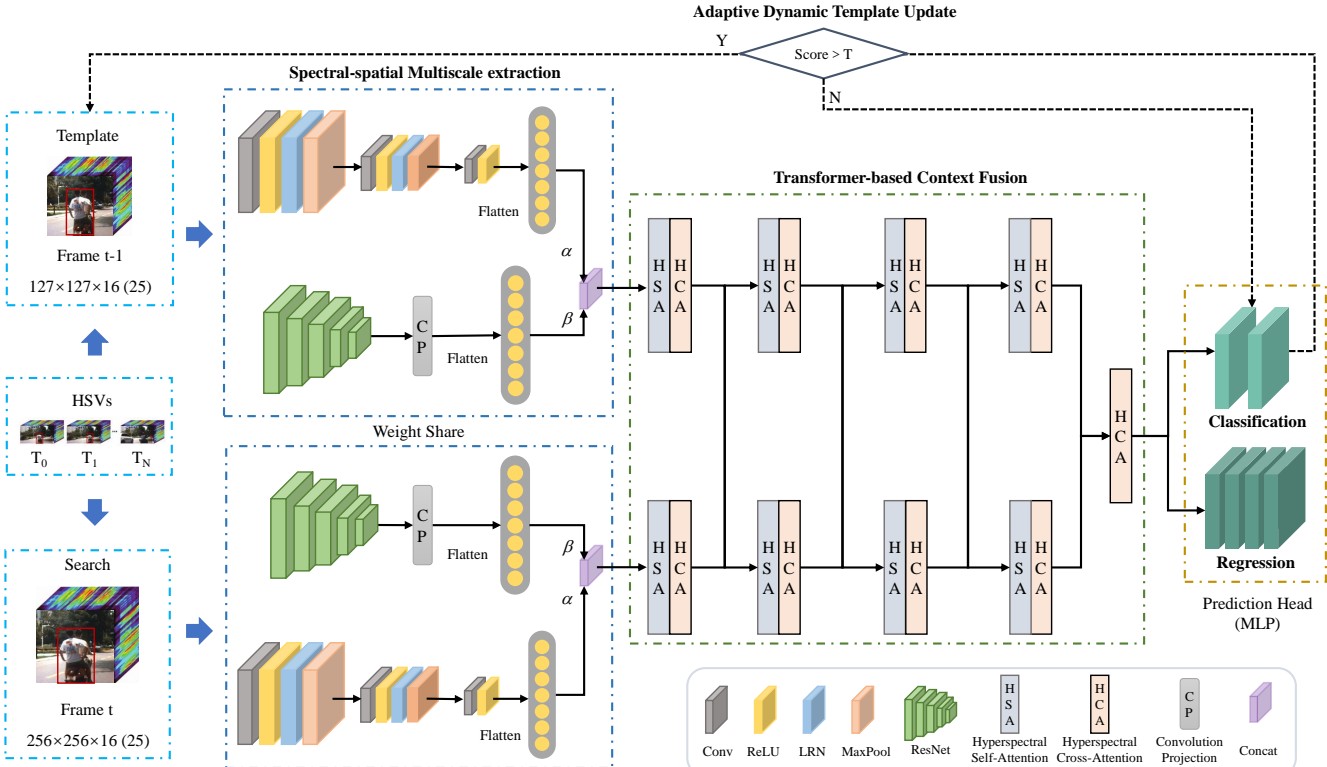

**Figure 1.** Illustration of the whole structure of the proposed hyperspectral video tracking, named SSTFT. The network extracts spectral–spatial multiscale features of the template and search region through the two branches. Specifically, these features interact with their own branches and are fused within the template and search branches using a transformer-based context fusion module to predict the bounding box and classification label. Finaly, the adaptive template update strategy is proposed to adapt the model to different challenges in real-time.

### 2.1. Spectral–Spatial Multiscale Feature Extraction Module

As is already known, spectral–spatial feature extraction is the most important procedure for hyperspectral videos. In order to save on computation cost, the template and search region branch share the same structures and weights, which are designed for high-dimensional spectral bands. In Figure 1, the Spectral–Spatial Multiscale Feature Extraction Module is shown to be composed of two parts in each branch, including the ResNet network and the spectral learning network, which are designed to extract the discriminative spectral–spatial features. The deep ResNet is capable of acquiring both shallow appearance information and deep semantic information, and fine spectral–spatial representations are integrated with this deep network. To enhance spectral information, the spectral learning network is adopted to utilize the original hyperspectral information and obtain more coarse spectral features. By employing this multiscale feature-extraction structure, the network can acquire both coarse-grained and fine-grained spectral–spatial information.

Formally, given the input hyperspectral video $X_T^t$ and $X_S^t$, which are the template and search region, respectively, the shallow spectral–spatial feature of the template branch in the $t$-th frame can be extracted as follows:

$$F_{T_{shallow}}^t = w_t \odot X_T^t + b_t \tag{3}$$

where $w_t$ and $b_t$ are the weight and bias of the shallow spectral–spatial feature extraction network, and $\odot$ is the element-wise multiplication.

Inspired by [37], the other deep spectral–spatial feature of the template branch in the $t$-th frame is extracted by ResNet, designed as follows:

$$F_{T_{deep}}^t = \mathcal{F}(X_T^t, \{W_i\}) + W_s X_T^t \tag{4}$$

where $\mathcal{F}$ is a residual function with multiple convolutional layers, $W_i$ is the weight matrix matching dimensions in the $i$-th layer, $W_s$ is a square matrix, and $+$ is the element-wise addition, which is performed on two feature maps channel by channel.

Finally, the features from two sub-branches are concatenated with the adaptive coefficient $\alpha$ and $\beta$ to obtain the spectral–spatial multiscale feature of the template branch in the $t$-th frame, designed as follows:

$$F_T^t = \alpha F_{T_{shallow}}^t + \beta F_{T_{deep}}^t \tag{5}$$

It is noted that the spectral–spatial multiscale feature of the search branch in the $t$-th frame operates in the same way as the template branch. Meanwhile, the two branches are injected with the inductive bias to avoid the overfitting problem, and the multiscale feature-extraction and shallow–deep subbranch modules are designed to extract the discriminative and robust features.

### 2.2. Transformer-Based Context Fusion Module

The transformer-based context fusion module, including Hyperspectral Self-Attention (HSA) and Hyperspectral Cross-Attention (HCA) in Figure 1, is designed to fuse the spectral–spatial features from the template branch and search branch with the transformer.

The HSA is utilized to realize the feature interaction of the context of each branch in Figure 2a. The HSA operation is the same for both the template branch and the search branch. Assume that $F_{SAT}^n$ and $F_{SAS}^n$ are the input spectral–spatial features of the template branch and search branch, respectively. Without loss of generality, the template branch is taken as an example in the left of Figure 2a, using $F_{SAT}^{n-1}$ as the input of the $n$-th HSA layer. The output of the previous layer is the input of the next layer, and the one head of the $n$-th HSA operation is designed as follows:

$$\begin{aligned} F_{SATh}^n &= LN\left(Softmax\left(\frac{(F_{SAT}^{n-1} + P_{SAT}^n)(F_{SAT}^{n-1} + P_{SAT}^n)^T}{\sqrt{d}}\right)F_{SAT}^{n-1}\right) \\ &= LN\left(Softmax\left(\frac{(Q_{SAT}^n)(K_{SAT}^n)^T}{\sqrt{d}}\right)V_{SAT}^n\right) \end{aligned} \tag{6}$$

where $Q_{SAT}^n = K_{SAT}^n = F_{SAT}^{n-1} + P_{SAT}^n$, $F_{SATh}^n$ is the output of template branch of the self-attention operation in the $h$-th head of the $n$-th layer, $h = 1, \ldots, H$, $H$ is the number of the heads, $n = 1, \ldots, 4$, $P_{SAT}^n$ is the absolute position embedding, $d$ is the dimension of the feature, and LN is the layer normalization. Then, the multiple heads are concatenated to obtain the $n$-th layer output of the self-attention operation, designed as follows:

$$F_{SAT}^n = Concat(F_{SAT1}^n, F_{SAT2}^n, \ldots, F_{SATH}^n) + F_{SAT}^{n-1} \tag{7}$$

where $F_{SAT1}^n, F_{SAT2}^n, \ldots, F_{SATH}^n$ are the output of the self-attention operation about multiple heads, and $+$ is the element-wise addition. The same self-attention operation is performed on the search branch to obtain the fusion feature on the right of Figure 2a.

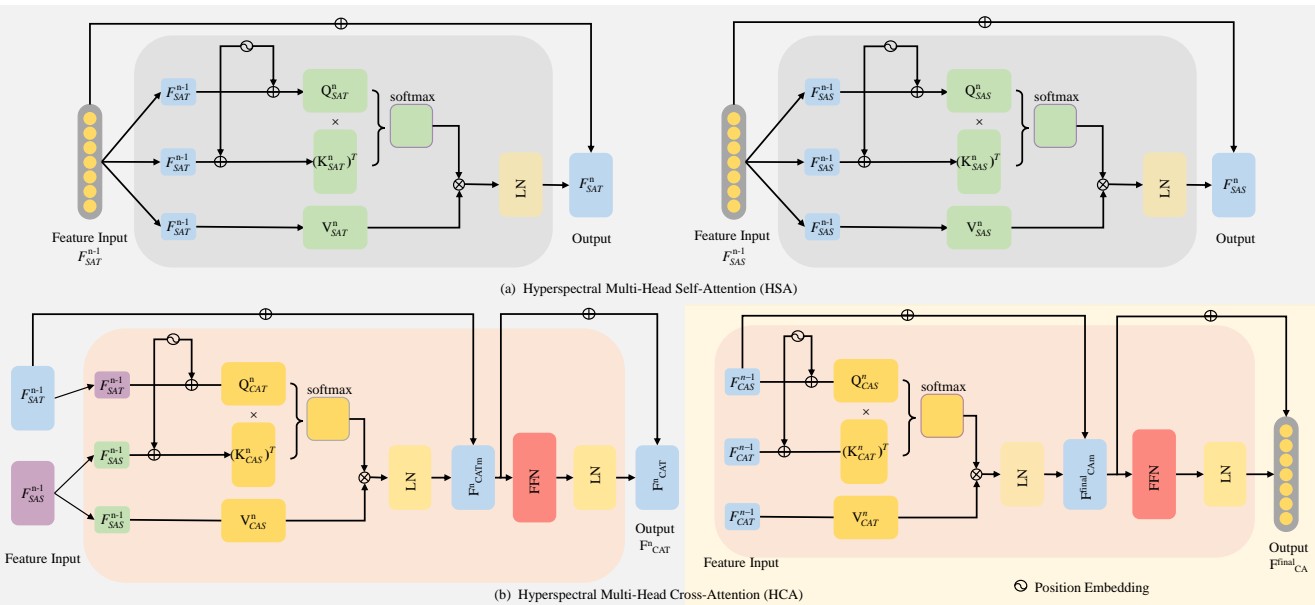

**Figure 2.** The detailed demonstrated ion of transformer-based context fusion. (**a**) The self-attention of the template branch in the *n*-th layer is displayed on the left, and that of the search region branch is on the right. (**b**) The cross-attention of the template branch in the *n*-th layer is displayed on the left, and the final cross-attention layer is on the right.

Meanwhile, the HCA, different from the HSA, conducts the hyperspectral cross-attention operation of their context and the feature interaction between the template branch and search branch in Figure 2b. Similarly to self-attention, without loss of generality, the template branch is taken as an example to introduce cross-attention. As shown on the left of Figure 2b, the common cross-attention operation in the *n*-th fusion layer, employed in the template branch, is designed as follows:

$$
\begin{aligned}
F_{CATh}^n &= LN(Softmax(\frac{(F_{SAT}^{n-1}+P_{CAT}^n)(F_{SAS}^{n-1}+P_{CAT}^n)^T}{\sqrt{d}})F_{SAT}^{n-1}) \\
&= LN(Softmax(\frac{(Q_{CAT}^n)(K_{CAS}^n)^T}{\sqrt{d}})V_{CAS}^n)
\end{aligned}
\tag{8}
$$

where $Q_{CAT}^n = F_{SAT}^{n-1} + P_{CAT}^n$, $K_{CAS}^n = F_{SAS}^{n-1} + P_{CAT}^n$, $F_{CATh}^n$ is the output of the cross-attention operation about one head, $h = 1, \ldots, H$, $H$ is the number of the heads, $n = 1, \ldots, 4$, $P_{CAT}^n$ is the absolute position embedding, $d$ is the dimension of the feature, and LN is the layer normalization. Then the multiple heads are concatenated to obtain the *n*-th layer output of the cross-attention operation, designed as follows:

$$
F_{CAT}^n{}' = Concat(F_{CAT1}^n, F_{CAT2}^n, \ldots, F_{CATH}^n) + F_{SAT}^{n-1}
\tag{9}
$$

where $F_{CAT1}, F_{CAT2}, \ldots, F_{CATH}$ are the output of the cross-attention operation about multiple heads and + is the element-wise addition. The final first cross-attention layer is designed as follows:

$$
F_{CAT}^n = F_{CAT}^n{}' + LN(FFN(F_{CAT}^n{}'))
\tag{10}
$$

where $F_{CAT}^n{}'$ is the output of the cross-attention operation for multiple heads in the template branch, $LN$ is the layer normalization, and $FFN$ is the feed-forward network. In addition, the FFN module is designed with two linear layers with a ReLU function in between, that is:

$$
FFN(x) = ReLU(xW_1 + b_1)W_2 + b_2
\tag{11}
$$

where $W_1$, $W_2$, $b_1$, and $b_2$ are the weight matrices and basis vectors in the FFN module, respectively. Furthermore, the different subscripts 1 and 2 represent the first and second linear layers, respectively.

Finally, when $n = 5$ in this paper, the spectral–spatial features of the two branches are fused with the last cross-attention layer. As shown in the right of Figure 2b, which is designed as follows:

$$
\begin{aligned}
F_{CAMh}^n &= LN(Softmax(\frac{(F_{CAS}^{n-1} + P^f)(F_{CAT}^{n-1} + P^f)^T}{\sqrt{d}})F_{CAT}^{n-1}) \\
&= LN(Softmax(\frac{(Q_{CAS}^n)(K_{CAT}^n)^T}{\sqrt{d}})V_{CAT}^n) \\
F_{CAm}^{final} &= Concat(F_{CAM1}^n, F_{CAM2}^n, \ldots, F_{CAMH}^n) \\
F_{CA}^{final} &= F_{CAm}^{final} + LN(FFN(F_{CAm}^{final}))
\end{aligned}
\tag{12}
$$

where $Q_{CAS}^n = F_{CAS}^n + P^f$, $K_{CAT}^n = F_{CAT}^{n-1} + P^f$, $F_{CAMh}^n$ is also the output of the cross-attention operation about one head, $P^f$ is the position embedding, $F_{CAm}^{final}$ is the concatenation of all heads in the final cross attention layer, and $F_{CA}^{final}$ is the final output. It must be emphasized that the feature is the output of the respective branches, which facilitates hyperspectral feature interaction in a spatio-temporal context. Furthermore, the search branch is also involved in the cross-attention operation as the template, which can effectively learn relationships between contexts.

The strong attention response map is shown in Figure 3, which is obtained by the last cross-attention layer. The maps contain less clutter than other weak response maps to achieve more accurate positioning. The center of the object is located at the position with the maximum value of the response map.

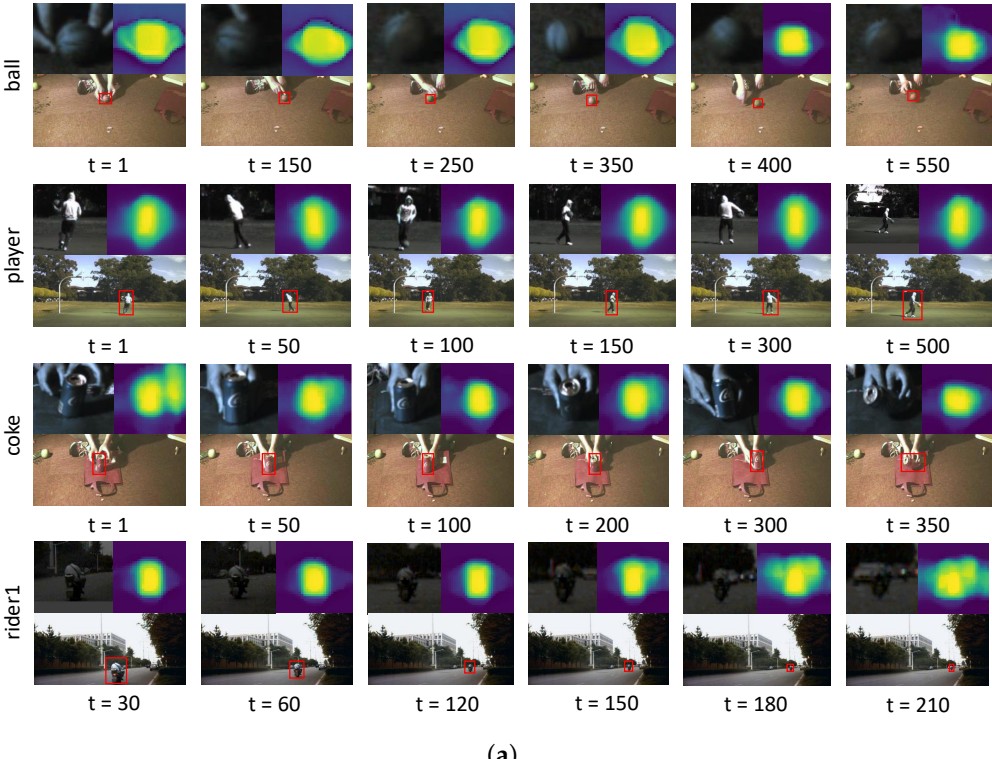

(**a**)

**Figure 3.** *Cont.*

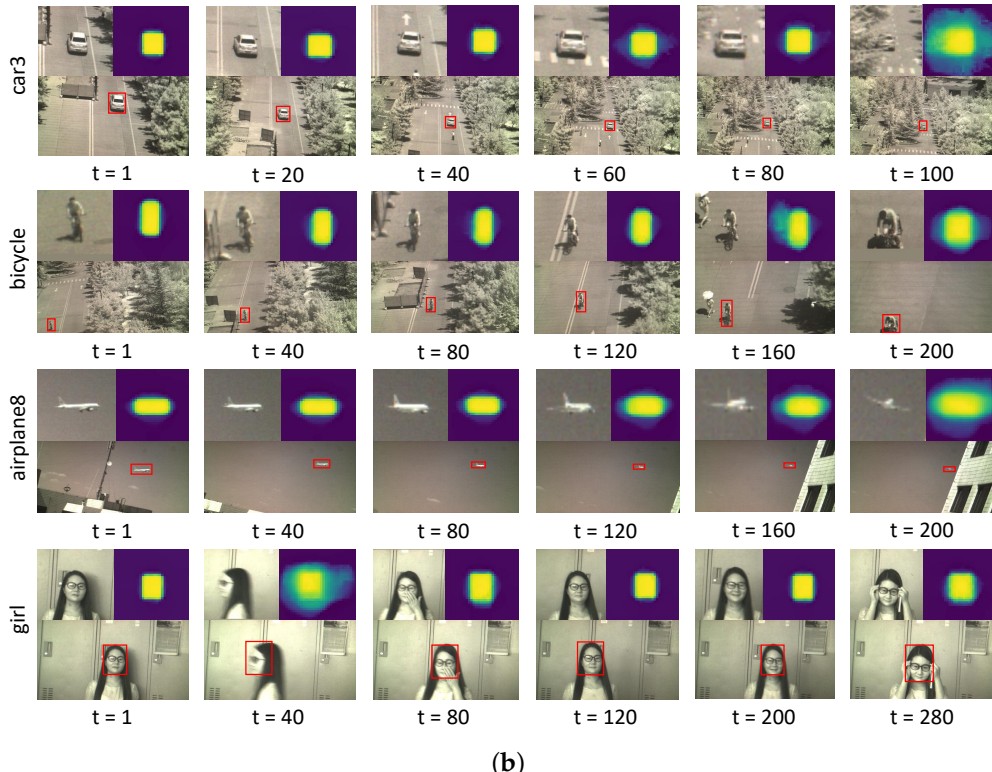

**Figure 3.** The strong attention response map for two datasets. It displays the attention map of objects in challenging scenarios. Each image consists of an initial image, cropped image, and the corresponding attention map. (**a**) The attention map of four sequences on IMEC16; (**b**) the attention map of four sequences on IMEC25.

### 2.3. Loss Function

The prediction head receives fusion feature vectors to output a binary classification and regression results. The positive sample is the feature vector corresponding to the pixel in the ground-truth bounding box, while the negative sample is the remainder. The whole sample contributes to the classification loss, while there is an imbalance problem between the positive sample and the negative sample, which is solved by down-weighing the loss of the negative sample with a factor of 16. The traditional binary cross-entropy loss in classification is defined as follows:

$$\mathcal{L}_{cls} = -\sum_t [y_t \log(p_t) + (1 - y_t) \log(1 - p_t)] \tag{13}$$

where $y_t$ is the ground-truth label in the $t$-th frame, $p_t$ is the predicted probability of object in the $t$-th frame, $p \in \{0, 1\}$, and $p_t = 1$ represents the foreground.

In addition, the regression loss is utilized as the positive sample, defined as follows:

$$\mathcal{L}_{res} = -\sum_t [\mathcal{L}_{IoU}(b_t, b_g) + \mathcal{L}_2(b_t, b_g)] \tag{14}$$

where $b_t$ is the predicted bounding box in the $t$-th frame, $b_g$ is the ground-truth bounding box, and $\mathcal{L}_{IoU}$ and $\mathcal{L}_2$ are the IoU loss and L2 loss, respectively.

### 2.4. Adaptive Dynamic Template Update Strategy

In previous Siamese tracking networks, it was common to use either a fixed time interval or response scores to update the template according to the last tracking result. However, these approaches may not be suitable for all hyperspectral video situations, such as that with object occlusion or fast motion. To address this problem, an adaptive template update strategy is proposed to dynamically adjust the threshold controlling the template

update. The threshold is not only determined by the last tracking frame but also affected by accumulation of all previous tracking scores, which can be described as follows:

$$\theta_t = \eta\theta_{t-1} + (1 - \eta)\mathbf{s}_t, \tag{15}$$

where $\theta_t$ is the accumulation threshold computed progressively in frames $1, 2, \ldots, t-1$ and $\theta_t$ is initialized as 0, $\eta$ is a momentum factor to control the influence of the previous threshold, and $\mathbf{s}_t$ is the tracking score of the current $t$-th frame.

It can be seen through Equation (15) that the threshold increases if the current tracking score is larger than the accumulation threshold; otherwise, the threshold decreases. The template is updated when tracking score $\mathbf{s}_t$ is greater than the current threshold. With this adaptive template update strategy, the tracking net is more likely to keep the template when the target is occluded or in fast motion, rather than updating the template with a low score, which can be seen as a more robust strategy.

## 3. Results and Analysis

In this section, the experimental results and analysis of the proposed method are reported. The details of the experimental settings are presented in Section 3.1, the public hyperspectral datasets are introduced in Section 3.2, and the comprehensive comparisons with the existing algorithms are presented in Sections 3.3–3.5. Moreover, to verify the advantages of the components of the proposed hyperspectral trackers mentioned, comparisons with ablation experiments are shown in Section 3.6.

### 3.1. Experiment Settings
### 3.1.1. Implementation Details

The proposed SSTFT method was implemented in PyTorch (version 1.5.1) and trained on NVIDIA GeForce RTX 2080 Ti GPU, which was trained using the Adam optimizer. The base learning rate was set to 0.0001, which was an adaptive modification following the poly learning rate policy with a power of 0.005. The batch size was set to 8, and the weight decay is set to 0.0001. The training process was terminated after 20 epochs. In the first 15 epochs, the fusion layers were frozen, and in the last 5 epochs, the fusion layers were trained. Specifically, our proposed SSTFT model was trained with pre-trained weights of the network [1] to resist the overfitting problem due to the shortage of samples, which provides an available reference initialization for the hyperspectral tracking method.

### 3.1.2. Evaluation Metrics

Five evaluation metrics are utilized to evaluate the performance of the SSTFT method—the area under the curve (AUC), location precision (LP), success score plot, precision score plot, and speed—to ensure fairness of comparison. All experimental results are reported on one-pass evaluation (OPE), and the whole trackers employed the test datasets. The AUC is the most commonly used evaluation metric for object tracking, and it is calculated as the average overlap rate when the threshold is less than 0.5. The LP is the percentage of the frame of Euclidean distance between the central location of the predicted and ground-truth bounding box at a threshold of 20 pixels (DP@20P). The success plot denotes the percentages of successful frames whose predicted bounding box and ground-truth overlap ratio is greater than a threshold varying from 0 to 1. A precision plot records the percentage of video frames whose distance between the center point of the estimated object position and the center point of the ground truth is less than a given threshold from 0 to 50 pixels. The speed is the average processing time in each frame.

### 3.2. Hyperspectral Datasets

The proposed method is evaluated on two public hyperspectral datasets, including IMEC16 and IMEC25. The details of the two datasets are shown in Sections 3.2.1 and 3.2.2.

### 3.2.1. IMEC16 Dataset

The IMEC16 dataset is a hyperspectral dataset collected by the hyperspectral camera of the IMEC company in [15]. The dataset is divided into 75 sequences, including 40 training sequences and 35 testing sequences. Each hyperspectral video sequence contains 16 spectral bands in wavelengths from 470 m to 620 m, and the spatial resolution is $512 \times 256$ pixels, as shown in Table 1. The entire dataset is manually labeled with bounding boxes representing the object by the central location and its height and width for each frame. The collected hyperspectral video sequences accompany the challenges, such as occlusion (OCC), illumination variation (IV), background clutter (BC), low resolution (LR), out-of-view (OV), in-plane rotation (IPR), out-of-plane rotation (OPR), deformation (DEF), motion blur (MB), scale variation (SV), and fast motion (FM). Meanwhile, the tracked objects include faces, pedestrians, animals, vehicles, etc. It is worth noting that the dataset involves similar objects in different scenes and the identical color of the target and background in different scenarios.

**Table 1.** IMEC16 Dataset Statistic Analysis for Close-range Object tracking.

| Classes | Sequences | Challenges | Classes | Sequences | Challenges |
|---|---|---|---|---|---|
| ball | 625 | SC, MB, OCC, SV | basketball | 186 | FM, MB, OCC, LR |
| board | 471 | IPR, OPR, BC, OCC, SV | book | 601 | IPR, DEF, OPR |
| bus | 131 | LR, BC, FM | bus2 | 326 | IV, SV, OCC, FM |
| campus | 970 | IV, SV, OCC | car | 101 | SV, OCC, IPR, OPR |
| car2 | 131 | SV, IPR, OPR | car3 | 331 | SV, LR, OCC, IV |
| card | 930 | IPR, BC, OCC | coin | 149 | BC |
| coke | 731 | BC, IPR, OPR, FM, SV | drive | 725 | BC, IPR, OPR, SV |
| excavator | 501 | IPR, OPR, SV, OCC, DEF | face | 279 | IPR, OPR, SV, MB |
| face2 | 1111 | IPR, OPR, SV, OCC | forest | 530 | BC, OCC |
| forest2 | 363 | BC, OCC | fruit | 552 | BC, OCC |
| hand | 184 | BC, SV, DEF, OPR | kangaroo | 117 | BC, SV, DEF, OPR, MB |
| paper | 278 | IPR, BC | pedestrain | 306 | IV, SV |
| pedestrain2 | 363 | OCC, LR, DEF, IV | player | 901 | IPR, DEF, OPR, SV |
| playground | 800 | SV, OCC | rider1 | 336 | LR, OCC, IV, SV |
| rider2 | 210 | LR, OCC, IV, SV | rubik | 526 | DEF, IPR, OPR |
| student | 396 | IV, SV | toy1 | 376 | BC, OCC |
| toy2 | 601 | BC, OCC, SV, IV, OPR | truck | 221 | OCC, IV, SV, OV |
| worker | 1209 | SV, LR, BC | | | |

### 3.2.2. IMEC25 Dataset

The other IMEC25 hyperspectral dataset was collected by a snapshot mosaic hyperspectral camera of the IMEC company in [13]. Chen et al., collected the hyperspectral surveillance video sequences with 25 spectral bands in wavelengths from 680 nm to 960 nm, and the spatial resolution was $409 \times 216$ pixels. The hyperspectral video dataset includes 135 manual sequences with 55 training sequences and 80 testing sequences. Each hyperspectral frame is also manually annotated with a bounding box that represents the

object by the central location and its height and width. The collected hyperspectral video sequences accompany the challenges that are different from the previous dataset, such as scale variation (SV), motion blur (MB), occlusion (OCC), fast motion (FM), background clutters (BC), low resolution (LR), in-plane rotation (IPR), out-of-plane rotation (OPR), deformation (DEF), fast motion (FM), illumination variation (IV), and out-of-view (OV). The average length of the hyperspectral video sequences is 174 frames, and the acquisition speed is 10 frames per second. The dataset is obtained in three typical real-world scenarios, including navigation, traffic, and take-off of the plane, in which tracked objects include ships, electric cars, pedestrians, bicycles, vehicles, and airplanes. It is worth noting that the tracked object is mostly small compared with the background with a motion shot, which brings more challenges for the hyperspectral tracking.

### 3.3. Quantitative Comparison

3.3.1. Quantitative Comparison with the Correlation Filters Tracking Methods

In this subsection, we reported the quantitative comparison of the proposed SSTFT method with the state-of-the-art Correlation Filter Trackers, including hyperspectral tracking methods and traditional CF algorithms. The comparison results of the IMEC16 and IMEC25 datasets are reported as Tables 2 and 3. The best values are highlighted in bold black, while an underscore is added to the second value. The same operation is adopted in the following sections.

**Table 2.** Comparison with the state-of-the-art correlation filters' trackers on IMEC16.

| Trackers | MHT | SSHMG | KCF | DCF | BACF | MCCT | fDSST | Struck | SAMF | STRCF | Ours |
|---|---|---|---|---|---|---|---|---|---|---|---|
| AUC | 0.587 | 0.578 | 0.352 | 0.310 | 0.561 | 0.539 | 0.482 | 0.357 | 0.416 | 0.456 | **0.682** |
| LP@(20) | 0.882 | 0.875 | 0.591 | 0.544 | 0.861 | 0.838 | 0.791 | 0.635 | 0.671 | 0.717 | **0.884** |

**Table 3.** Comparison with the state-of-the-art correlation filters trackers on IMEC25.

| Trackers | CSRDCF | CSK | HOMG | ECO | BACF | MCCT | DSST | LDES | Ours |
|---|---|---|---|---|---|---|---|---|---|
| AUC | 0.527 | 0.222 | **0.746** | 0.598 | 0.540 | 0.552 | 0.502 | 0.449 | 0.619 |
| LP@(20) | 0.775 | 0.429 | 0.823 | 0.634 | 0.797 | 0.797 | 0.720 | 0.650 | **0.888** |

As shown in Table 2, the proposed SSTFT method achieves the top performance on the IMEC16 dataset among the evaluated 10 trackers including MTH [15], SSHMG [15], KCF [38], BACF [39], MCCT [40], fDSST [41], Struck [42], SAMF [43], and STRCF [44].

Each tracker is briefly introduced as follows. The KCF tracker is a fast kernelized correlation filter to capture non-linear classification boundaries. The MCCT tracker constructs the divergence of multiple experts through a discriminative correlation filter, in which a suitable expert is selected for tracking adaptively in the current frame. The fDSST tracker works to learn discriminative correlation filters based on a scale pyramid representation, which separates filters for translation and scale estimation. The Struck tracker adopts a kernelized structured support vector machine (SVM) to predict the object location by online tracking. The SAMF tracker extracts HoG [45] and color-naming features with a scale-adaptive scheme. The STRCF tracker is based on spatial–temporal correlation filters to handle boundary effects, which provides reasonable approximation with multiple training samples. The BACF tracker is a background-aware CF based on hand-crafted features such as HoG, which is designed to distinguish the foreground and background of the object. The SSHMG is the hyperspectral feature descriptor based on spectral–spatial structure information, which is the fundamental feature to represent the local material in the hyperspectral image. The MTH is also a hyperspectral tracker based on the BACF tracker while considering the material and fractional abundance information to recognize the object in complex scenarios. The spectral–spatial histogram of multidimensional gradients is a fundamental feature to represent the material of the object in the hyperspectral image,

while fractional abundances encode the underlying material distribution. Both feature descriptors capture the local spectral and spatial information of the object.

As shown in Table 2, the SSTFT, SSHMG, and MHT trackers are tested on the hyperspectral dataset, while the other trackers are tested on the false-color videos generated from hyperspectral images. The results show that the KCF, DCF, and Struck obtained unsatisfying AUC and LP scores, which are due to ignoring scale estimation. BACF, MCCT, SAMF, and STRCF trackers are based on learning more discriminative feature descriptors, leading to much better performance. The fDSST tracker presents better performance than the KCF and DCF trackers, which is due to the scale estimation. The MTH and SSHMG trackers are based on hyperspectral feature descriptors, which obtain better performance than the other compared trackers. It is worth mentioning that the proposed SSTFT tracker achieves higher performance over the MHT, providing a gain of 9.5% and 0.2% in AUC and LP, respectively. Meanwhile, compared with other CF trackers, the proposed SSTFT tracker achieves the top performance on the IMEC16 dataset by obtaining 68.2% in AUC and 88.4% in LP. This implies that the proposed tracker has the ability to distinguish the object from the background using the spectral–spatial information and global feature distribution information contained in the hyperspectral video. The SSTF tracker is adapted to complex scenarios, which is beneficial to hyperspectral video tracking.

As shown in Table 3, the proposed SSTFT method ranks the first on the IMEC25 dataset among the eight evaluated trackers, i.e., CSRDCF [46], CSK [47], HOMG [25], ECO [48], BACF [39], MCCT [40], DSST [49], and LDES [50]. The CSRDCF tracker introduces the channel and spatial reliability score to the discriminative correlation filter tracker, which utilizes HoGs features and Colornames features. The CSK tracker extends ridge regression and an approximate dense sampling method based on cyclic shift. The HOMG tracker is a hyperspectral object-tracking method based on the histograms of an oriented mosaic gradients descriptor. The ECO tracker is a fast, compact generative model based on a factorized convolution operator in which there is a trade-off between speed and accuracy. The LDES tracker is a correlation-filter-based tracker with a robust estimation to simultaneously handle changes in both scale and rotation. The results show that the CSK, which is the single-channel grayscale tracker, presents the poorest performance. In contrast, the CSRDCF and ECO tracker obtains improved performance due to the fact that diverse types of features are combined to learn the discriminative characteristics, while it is not very suitable for hyperspectral video tracking. The HOMG tracker focuses on the hyperspectral mosaic gradient features, which are employed to distinguish the object from the background. The DSST tracker reduces all the scale detection image blocks to the same size to compute feature (CN+HoG) and then represents the feature as one dimension. However, the tracker tends to obtain the local spectral–spatial information and does not achieve the best positioning accuracy. The other CF trackers are based on hand-crafted features, which are not suitable for hyperspectral video tracking. The SSTFT method, achieving the top LP score (0.888) and the second highest AUC score (0.619), can distinguish the object from the background using the spectral–spatial information and global feature distribution information contained in the hyperspectral video. This implies that the proposed tracker is adapted to complex scenarios, which is beneficial for hyperspectral video tracking.

### 3.3.2. Quantitative Comparison with State-of-the-Art Deep Learning Tracking Methods

In this section, we compare the proposed SSTFT tracker with the state-of-the-art deep learning trackers on the IMEC16 in Figure 4 and IMEC25 in Figure 5. Except for the proposed method and BAENet, the other deep-learning trackers were tested on the false-color videos generated from hyperspectral images.

The proposed SSTFT tracker was compared with nine trackers, namely SiamFC [51], SiamRPN [52], SiamRPN++ [53], DaSiamRPN [54], SiamBAN [55], MDNet [56], BAENet [57], SiamHYPER [23], and TransT [1], on the IMEC16 dataset. The SiamFC tracker is the first Siamese deep-learning tracker, and it is based on a fully convolutional Siamese network and combines the resulting feature maps based on the cross-correlation layer. The

SiamRPN tracker introduces the region proposal network to generate the classification label and regression boxes for tracking. The SiamRPN++ tracker provides a simple yet effective spatially aware sampling strategy in which multi-level features are extracted from the residual block for layer-wise aggregation to improve the performance of the SiamRPN tracker. The DaSiamRPN tracker further explores a distractor-aware module to improve the localization accuracy of the model. The SiamBAN tracker consists of the expressive power of the fully convolutional network, which does not require pre-defined candidate boxes or multi-scale searching. The MDNet tracker consists of shared layers and multiple branches of domain-specific layers, where generic object representations are obtained in the former layers, and domain-specific representations are obtained in the latter layers. The BAE-Net is based on that of Vital [58] to explore the spectral–spatial feature of the hyperspectral video, which predicates the tracking results based on ensemble learning. The TransT tracker is introduced to learn the template and search region feature solely using attention. The SiamHYPER is a multi-modal hyperspectral tracker combined with a pre-trained RGB tracker and a spatial–spectral cross-attention module to be aware of the location of the object.

As shown in Figure 4, the SSTFT method achieves the top performance on the IMEC16 dataset by achieving 88.4% in the precision plot score, the second performance by achieving 68.2% in success plot score, and 82.5% in normalization precision plot score. Compared with the hyperspectral tracker BAENet, the proposed tracker gains improvements with 7.5% in the success plot score, 0.6% in the precision plot score, and 5.8% in the normalization precision plot score. The reason for the lower performance of BAENet is that several spectral information is ignored in the band attention module. The reason why other deep learning trackers present lower success rate scores and precision rate scores is the loss of useful spectral information in the false-color videos for the challenging scenarios of the dataset. The most important reason that the SiamHYPER obtained the top precision rate score is the use of highly aligned multi-modal data with visible and hyperspectral information. However, such methods face a very demanding challenge of data sets, which requires a lot of storage resources and human resources.

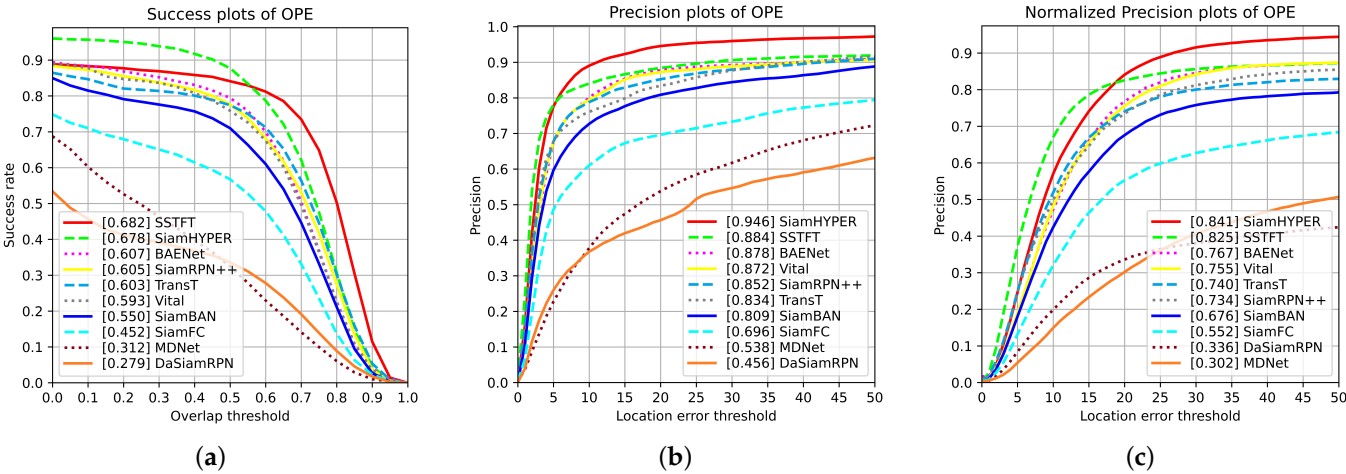

**Figure 4.** Comparisons with deep learning trackers on IMEC16.

The proposed SSTFT tracker is compared with six deep learning trackers, namely SiamFC [51], SiamRPN++ [53], DASiamRPN [59], UpdateNet [60], and TransT [1], on the IMEC25 dataset. The SiamCAR is a Siamese network consisting of a feature extraction subnetwork and a bounding box prediction subnetwork, which is based on a proposal and is anchor-free without tricky hyper-parameter tuning of boxes. The UpdateNet tracker introduces a learned update strategy with target and image information to update the template. The other compared trackers are the same as the IMEC16 dataset, and the corresponding results are implemented on three bands of video sequences.

The proposed tracker achieves the top performance on the IMEC25 dataset as shown in Figure 5, which is the first deep learning tracker based on hyperspectral video datasets with 25 bands, by achieving 88.8% in precision plot score, 61.9% in success plot score, and 77.9% in normalization precision plot score. It is worth noting that the proposed trackers obtain a performance improvement of 7.2% in success plot score, 1.7% in precision plot score, and 3.8% in normalization precision plot score compared with the transformer tracker named TransT. The traditional TransT is trained on optical datasets, and the performance of TransT is not satisfactory on the hyperspectral video dataset. DaSiamRPN and SiamCAR trackers obtain inferior performance in that the target is not easy to distinguish in challenging video surveillance scenarios without the use of abundant spectral information. The most important reason why the SSTFT tracker obtains the best performance on the IMEC25 and IMEC16 datasets is that the proposed tracker utilizes the spectral–spatial features of the hyperspectral video. Meanwhile, the feature global interaction is beneficial to predict the data distribution.

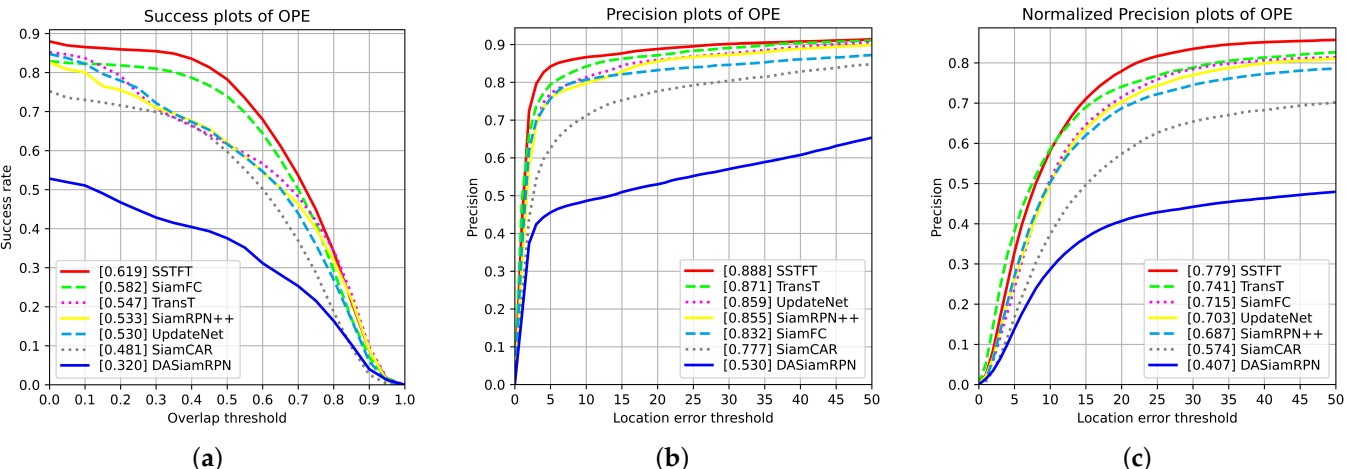

**Figure 5.** Comparisons with deep learning trackers on IMEC25.

The floating point of operations (FLOPs) and parameters are reported in Table 4, from which it can be obviously found that our method has the lowest computational complexity.

**Table 4.** Model complexity comparison.

| Trackers | FLOPs | Parameters |
|:---:|:---:|:---:|
| **SiamBAN** | 51.5 G | 53 M |
| **TMTNet** | 1846.6 G | 333.4 M |
| **Ours** | 3.36 G | 20 M |

### 3.3.3. Attribute-Based Evaluation

In this section, we report the effectiveness of the proposed SSTFT tracker on the public hyperspectral video datasets IMEC16 and IMEC25 with different attributes. The result of attribute-based evaluation is reported in Figures 6–8 and Tables 5 and 6.

**Table 5.** Comparison with the other trackers in terms of precision rate on IMEC25.

| Attributes | BACF | CSRDCF | UpdateN | MCCT | DSST-LP | SiamFC | DA-SiamRPN | SiamCAR | Siam-RPN++ | TransT | Ours |
|---|---|---|---|---|---|---|---|---|---|---|---|
| SV | 0.685 | 0.831 | 0.844 | 0.834 | 0.709 | 0.840 | 0.605 | 0.692 | <u>0.888</u> | **0.946** | 0.846 |
| MB | 0.841 | 0.816 | <u>0.914</u> | 0.852 | 0.740 | **0.944** | 0.657 | 0.813 | 0.876 | 0.847 | 0.890 |
| OCC | 0.810 | 0.750 | 0.851 | 0.760 | 0.664 | 0.823 | 0.452 | 0.726 | <u>0.863</u> | 0.843 | **0.889** |
| FM | 0.800 | 0.642 | <u>0.857</u> | 0.692 | 0.577 | 0.838 | 0.619 | 0.618 | 0.829 | 0.750 | **0.871** |
| BC | 0.737 | 0.702 | 0.741 | <u>0.816</u> | 0.742 | 0.776 | 0.607 | 0.659 | 0.753 | 0.802 | **0.836** |
| LR | 0.779 | 0.772 | 0.819 | <u>0.916</u> | 0.701 | 0.869 | 0.503 | 0.750 | 0.808 | 0.890 | **0.920** |
| IPR | 0.568 | 0.532 | 0.742 | 0.630 | 0.448 | 0.695 | 0.604 | 0.592 | 0.809 | <u>0.820</u> | **0.884** |
| OPR | 0.760 | 0.655 | <u>0.814</u> | 0.704 | 0.714 | 0.778 | 0.398 | 0.700 | 0.819 | 0.803 | **0.882** |
| DEF | 0.764 | 0.696 | 0.794 | 0.730 | 0.724 | 0.786 | 0.420 | 0.685 | 0.775 | <u>0.807</u> | **0.833** |
| IV | 0.840 | 0.791 | <u>0.898</u> | 0.742 | 0.795 | 0.802 | 0.592 | 0.819 | **0.924** | 0.795 | 0.852 |
| OV | 0.648 | 0.763 | **0.889** | 0.650 | 0.598 | 0.820 | 0.661 | 0.764 | <u>0.883</u> | 0.841 | 0.864 |

**Table 6.** Comparison with the other trackers in terms of success rate on IMEC25.

| Attributes | BACF | CSRDCF | UpdateNet | MCCT | DSST-LP | SiamFC | DA-SiamRPN | SiamCAR | Siam-RPN++ | TransT | Ours |
|---|---|---|---|---|---|---|---|---|---|---|---|
| SV | 0.397 | 0.530 | 0.569 | 0.526 | 0.463 | 0.591 | 0.393 | 0.420 | 0.592 | <u>0.609</u> | **0.619** |
| MB | 0.560 | 0.553 | 0.584 | 0.599 | 0.523 | **0.685** | 0.404 | 0.460 | 0.584 | 0.518 | <u>0.618</u> |
| OCC | 0.548 | 0.522 | 0.584 | 0.539 | 0.463 | 0.578 | 0.314 | 0.479 | <u>0.604</u> | 0.562 | **0.656** |
| FM | 0.578 | 0.462 | <u>0.622</u> | 0.519 | 0.429 | 0.602 | 0.432 | 0.410 | 0.608 | 0.515 | **0.635** |
| BC | 0.485 | 0.461 | 0.469 | <u>0.560</u> | 0.504 | 0.533 | 0.367 | 0.357 | 0.465 | 0.454 | **0.568** |
| LR | 0.499 | 0.447 | 0.311 | <u>0.555</u> | 0.416 | **0.567** | 0.188 | 0.380 | 0.295 | 0.420 | 0.526 |
| IPR | 0.421 | 0.396 | 0.501 | 0.456 | 0.348 | 0.512 | 0.395 | 0.374 | <u>0.557</u> | 0.536 | **0.657** |
| OPR | 0.530 | 0.459 | 0.599 | 0.506 | 0.512 | 0.564 | 0.278 | 0.467 | <u>0.618</u> | 0.557 | **0.666** |
| DEF | 0.501 | 0.478 | 0.532 | 0.507 | 0.508 | <u>0.543</u> | 0.261 | 0.436 | 0.525 | 0.511 | **0.576** |
| IV | 0.581 | 0.595 | 0.646 | 0.553 | 0.607 | 0.584 | 0.414 | 0.551 | **0.675** | 0.570 | <u>0.651</u> |
| OV | 0.487 | 0.545 | 0.627 | 0.465 | 0.445 | 0.604 | 0.467 | 0.491 | 0.651 | <u>0.681</u> | **0.703** |

As shown in Figures 6 and 7 on IMEC16, the proposed tracker is compared with sixteen trackers, including eight deep learning trackers and eight correlation filter trackers, in challenging scenarios such as BC, DEF, LR, and OV. In Figure 6, SSTFT achieves the top performance on BC, DEF, LR, and OV attributes in terms of the success plot score and precision plot score. It is worth noting that SSTFT shows outstanding results in LR and OV attributes, in which the object is partially out of snapshot or under low-resolution scenarios. In the LR attribute, compared with the tracker ranking second, SSTFT brings a gain of 13.7% in the success plot score and 5.1% in the precision plot score, while in the OV attribute, compared with the second best tracker, SSTFT brings a gain of 13.0% in success plot score and 3.2% in precision plot score. The reason may be that the spectral–spatial features of the hyperspectral video are fully utilized in the proposed method, and the features of global interaction are beneficial to predict data distribution to locate tracking objects. The other trackers based on the false-color images may not recognize the specific properties of the object from the complex background. In addition, as shown in Figure 7, various attributes are evaluated on the IMEC16 dataset with a radar graph. SSTFT achieves the most outstanding performance in robustness and accuracy from statistics of success rate and precision rate since the index of the proposed method is relatively dense in various attributes.

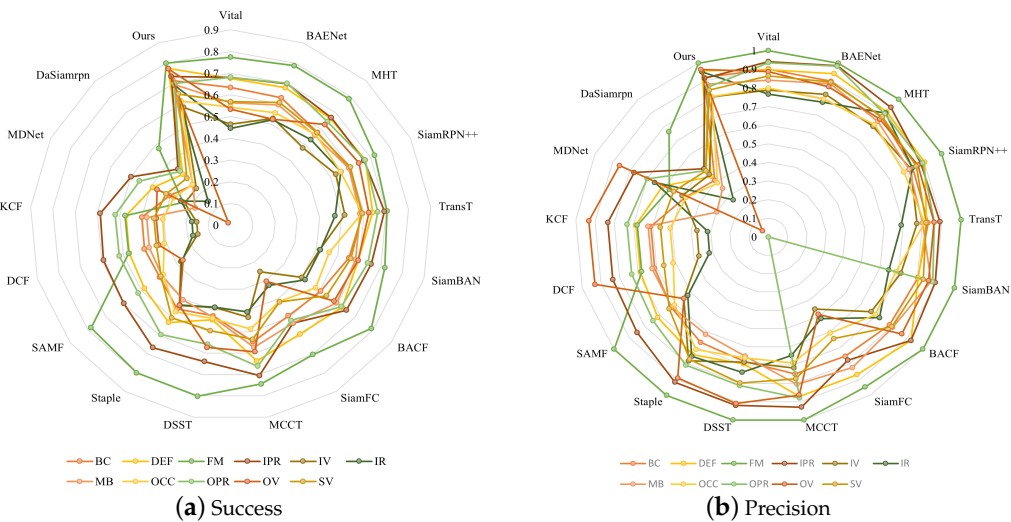

**Figure 6.** The proposed method compared with all trackers on IMEC16.

**Figure 7.** The proposed method compared with trackers on IMEC16.

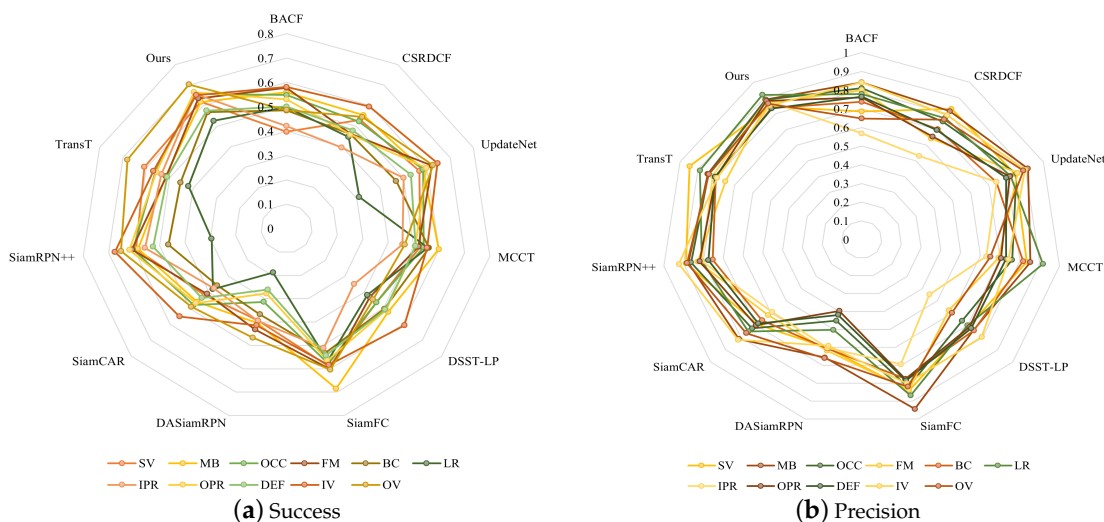

**(a)** Success       **(b)** Precision

**Figure 8.** The proposed method compared with trackers on IMEC25.

On the IMEC25 in Tables 5 and 6 and Figure 8, 11 attributes such as SV, MB, and OCC are evaluated. As shown in Table 5, SSTFT ranks first on 7 out of 11 attributes, including OCC, FM, BC, IPR, OPR, and DEF, in terms of precision rate. As shown in Table 6, the SSTFT ranks first on 8 out of 11 attributes, including SV, OCC, FM, BC, IPR, OPR, DEF, and OV, in terms of success rate. The challenging attribute BC results in great difficulty in extracting discriminative features from the object in the image. However, the hyperspectral cube with abundant spectral information can be effectively utilized to distinguish the object from the background in SSTFT. In addition, SSTFT displays an excellent precision rate in the OV attribute, which brings a gain of 2.2% in precision rate compared with TransT. Meanwhile, SSTFT achieves the top performance in the LR attribute, which brings a gain of 3.0% in success rate compared with TransT. The DEF is also one of the most challenging attributes, which is a target that has partly or fully deteriorated. However, SSTFT still yields satisfactory results in this attribute by achieving 0.576 and 0.833 in success rate and precision rate, respectively. The reason is that the spectral–spatial features of the hyperspectral video are fully explored in the proposed tracker, and the global interaction structure is beneficial to improve the performance of the model. Meanwhile, as shown in Figure 8, SSTFT achieves the most outstanding performance in robustness and accuracy from the statistics of success rate and precision rate. A possible reason is that the deep features consisting of the coarse-grained cube and fine-grained cube, underlying physical and robust attributes, are employed to capture the discriminative information of the object compared with other trackers based on hand-crafted features.

### 3.4. Qualitative Comparison

In this section, the qualitative evaluations of the competing trackers on hyperspectral or false-color videos are provided, as shown in Figures 9 and 10. Figure 9 reports the visual results of the proposed method and six compared trackers on six challenging sequences on IMEC16. For the sequence fruit, coke, toy1, and coin, the object is similar to the background, which makes it difficult to locate. At the initial stage (e.g., #174 in fruit, #306 in coke, and #153 in toy1), SSTFT, Vital, Staple, MHT, and BACF succeed in tracking to track the object, while SiamBAN fails to track the fruit, and TransT fails to track the toy1. In #275 of the fruit sequence, SSTFT shows the best performance, while other trackers have different degrees of drift. In #728 of the coke sequence, SSTFT has stable tracking performance, while SiamBAN and Vital display a slight drift, and BACF, MHT, and Staple lose the object. In addition, TransT predicts the wrong position and scale of the object from #554 to #728. In #233 and #270 of the toy1 sequence, other trackers such as SiamBAN, Staple, and MHT predict a similar false object, while SSTFT still maintains optimal and stable performance.

For the sequence of the forest and worker, the object suffers from BC challenge, which makes the tracker confused in locating the object. In contrast, SSTFT performs well in the whole sequence, which implies that it can handle challenging scenarios.

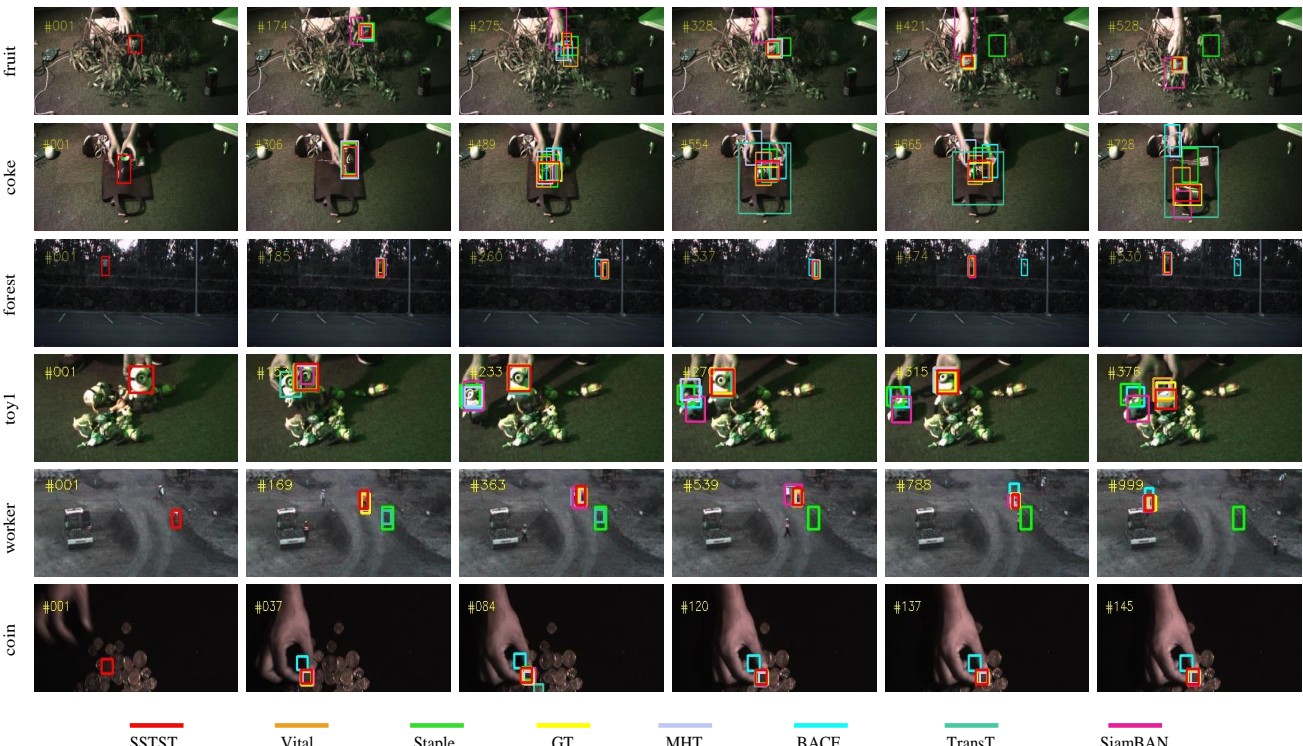

**Figure 9.** Qualitative results of the proposed method on some challenging sequences on IMEC16.

Figure 10 reports the visual results of the proposed method and seven compared trackers on six challenging sequences on IMEC25. It can be seen that the tracked object is small compared with the background in sequence man3, boat8, double8, doublecar8, and triple2. For sequence man3, the scale and appearance of the object change significantly because of the fast motion and out-of-plane rotation. Besides SSTFT, the other trackers do not perform well in the whole sequence. For sequences boat8, double8, and triple2, the object is too small to recognize from the background. Furthermore, similar target interference is also an enormous challenge for the tracker. In #155, #257, #323, and #461 of sequence boat8, SSTFT has the best tracking performance, while there are two types of tracking failures for the other trackers: one is seriously drifting, and the other is tracking similar targets. For sequences double8 and triple2, SSTFT is the only tracker that can predict the correct position and scale of the target. For sequence doublecar8, the tracked car is partially occluded by trees in #93 and #105, and the appearance of the car is significantly changed because of the deformation in #172 and #198.

Although other trackers can locate the object well in #32, the estimated object localization and scales are not accurate in a subsequent frame. In addition, for the airplane8 sequence, the airplane is moving fast, with the challenge of out-of-plane rotation. DaSiamRPN has a drift, whereas SSTFT can successfully track the airplane. Therefore, our SSTFT performs more robustly.

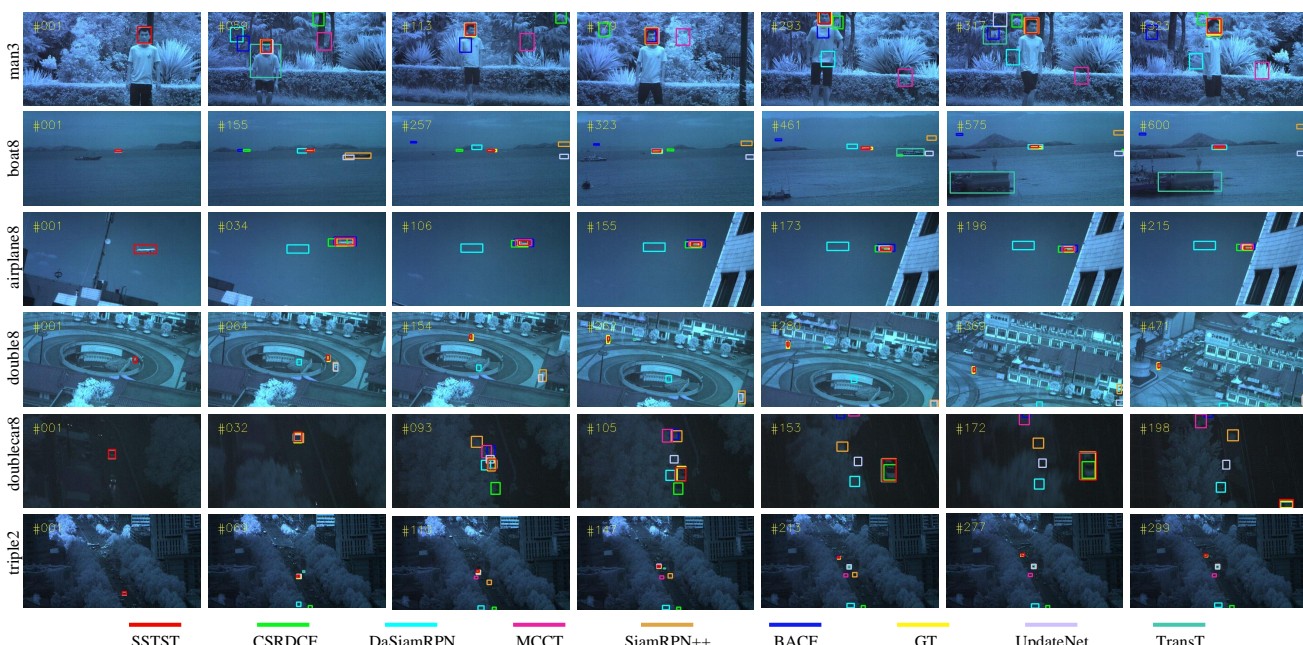

**Figure 10.** Qualitative results of the proposed method on some challenging sequences on IMEC25.

### 3.5. Running Time Analysis

Tables 7 and 8 report the running time of the proposed method and compare the trackers on the IMEC16 and IMEC25 datasets, which record the frames per second (*fps*) with identical hardware facility. As shown in Table 7, the running time of the proposed method is 19.9 *fps* on the IMEC16 dataset, which is much faster than MDNet, Vital, MCCT, and MHT. Due to the high-dimensionality features of MHT, SSTFT is 7.7 times faster than MHT. Furthermore, MDNet and Vital sacrificed speed for the accuracy by online training, which is 8.3 and 8.7 times slower than SSTFT. Although TransT, SiamBAN, and SiamFC are slightly faster than SSTFT, the performance of SSTFT is better than theirs. The reason is that the compared trackers ignore the spectral information, which is important for target tracking in hyperspectral video. However, as shown in Table 8, the speed of the proposed method is 20.4 *fps* on the IMEC25 dataset, which is much faster than HOMG and ECO. Compared with HOMG extracted from the mosaic spectral image, SSTFT is more than 9.3 times faster. Similarly, SSTFT is more than 15.7 times faster than ECO. The most important reason is the compared high-dimensionality features of trackers, which have a high computational cost. While the other compared trackers, named DaSiamRPN and SiamBAN, that are based on deep learning are much faster than SSTFT, the performance is not satisfactory due to the lack of spectral information. Therefore, the proposed method is verified to be effective and robust in hyperspectral video tracking, which is much faster than most compared trackers.

**Table 7.** Comparison the running time on IMEC16.

| Trackers | TransT | SiamBAN | SiamFC | MDNet | Vital | MCCT | MHT | STRCF | Ours |
|----------|--------|---------|--------|-------|-------|------|-----|-------|------|
| **Speed** | 22.2 | 24.4 | 10.9 | 2.4 | 2.3 | 8.2 | 2.6 | 20.3 | 19.9 |

**Table 8.** Comparison the running time on IMEC25.

| Trackers | TransT | SiamBAN | DaSiamRPN | SiamFC | BACF | HOMG | ECO | Ours |
|----------|--------|---------|-----------|--------|------|------|-----|------|
| **Speed** | 23.3 | 26.3 | 90.1 | 10.6 | 28.1 | 2.2 | 1.3 | 20.4 |

*3.6. Ablation Experiments*

In this section, we report the ablation experiments to verify the effectiveness of the proposed method. According to the experimental experience, the transformer obtained the best tracking performance when it had four fusion layers, and we therefore set the number of capacitive fusion layers to four. The result of the experiments is evaluated on the IMEC16 and IMEC25 datasets.

3.6.1. Ablation Experiments on IMEC16

The proposed SSTFT contains three components: a spectral–spatial multiscale feature extraction module, a context fusion module based on the transformer, and an adaptive dynamic template update strategy. In order to investigate the contributions of each component of SSTFT, ablation experiments were conducted on the IMEC16 and IMEC25 datasets according to three variants, SSTFT-noSSS, SSTFT-noDRN, and SSTFT-noADTU. SSTFT-noSSS represents the features without the shallow spectral–spatial feature extraction module. SSTFT-noDRN represents the features without the deep spectral–spatial feature extraction module. SSTFT-noADTU represents the features without the adaptive dynamic template update strategy.

Table 9 reports the results of the ablation study on the IMEC16 dataset. Each component of SSTFT positively affects the tracking performance. SSTFT achieved {0.682, 0.884} in AUC and DP, which ranks the first in all implementations, and obtained {4.6%, 0.6%}, {52.1%, 54.3%}, {26.9%, 32.7%} improvement in AUC and DP compared with SSTFT-noSSS, SSTFT-noDRN, and SSTFT-noADTU, respectively. The results show that the spectral–spatial multiscale feature extraction module, the transformer-based context fusion module, and the adaptive dynamic template update strategy are complementary to each other. Therefore, it is effective to construct a customized hyperspectral feature-extraction and data-interaction model for the object-tracking task.

**Table 9.** Ablation experiment on IMEC16.

| Trackers | SSTFT-noSSS | SSTFT-noDRN | SSTFT-noADTU | Ours |
|----------|-------------|-------------|--------------|------|
| AUC | 0.636 | 0.161 | 0.413 | **0.682** |
| DP | 0.878 | 0.341 | 0.557 | **0.884** |

3.6.2. Ablation Experiments on IMEC25

Table 10 reports the results of the ablation study on the IMEC25 dataset. The proposed three components of SSTFT also improve the performance in hyperspectral object tracking. SSTFT obtained the best performance, which is {0.619, 0.888} in AUC and DP. The final results show that SSTFT achieved an improvement of 10.1% in AUC, 8.3% in DP compared with SSTFT-noSSS, 38.3% in AUC, and 37.9% in DP compared with SSTFT-noDRN, yielding a gain of 32.2% in AUC, 45.3% in DP compared with SSTFT-noADTU. The method without deep ResNet provided the worst accuracy among the compared trackers, as the low dimensionality of the features could not represent the fine-grained information of the object. The method without the adaptive dynamic template update strategy also performed poorly, which is due to the fact that the fixed template leads to the model losing the ability to adapt to the object deformation. Therefore, the proposed method has shown effectiveness and robustness in the hyperspectral object tracking task.

**Table 10.** Ablation experiment on IMEC25.

| Trackers | SSTFT-noSSS | SSTFT-noDRN | SSTFT-noADTU | Ours |
|----------|-------------|-------------|--------------|------|
| AUC | 0.518 | 0.236 | 0.297 | **0.619** |
| DP | 0.805 | 0.509 | 0.435 | **0.888** |

## 4. Conclusions

This paper proposed a novel SSTFT model to extract the spatial–spectral feature from hyperspectral videos. The confluent features were extracted with the spatial–spectral

multiscale feature extraction module. Then, the context interaction module based on the transformer structure was designed to interact with the fused features between the template branch and the search branch. To evaluate the performance of the proposed SSTFT model, we integrated it into the challenging visual tracking task, and extensive experiments were performed on two datasets collected by different snapshot spectral cameras. Great experiment results illustrate the effectiveness and generalization of the proposed SSTFT in the obtained spatial–spectral features. Meanwhile, the research also provides a new perspective for hyperspectral visible tracking using the transformer structure. The proposed method has adequate ability to handle various complex challenges in hyperspectral object tracking.

**Author Contributions:** Y.W. and S.M. provided conceptualization; Y.W. and Y.L. explored related works and designed the methodology; Y.W. and Y.L. employed the experiments and analyzed the tracking results; Y. W. wrote the paper; S.M., M.M., and Y.L. put forward suggestions for revision of the manuscript. All authors have read and agreed to the published version of the manuscript.

**Funding:** This work is supported in part by the National Natural Science Foundation of China under Grant 62171381 and Grant 62201445, and in part by Innovation Foundation for Doctor Dissertation of Northwestern Polytechnical University under Grant CX2021080.

**Data Availability Statement:** Our experiments employ public datasets in [15,25].

**Conflicts of Interest:** The authors declare no conflict of interest.

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
