# Peer review of "A Spectral–Spatial Transformer Fusion Method for Hyperspectral Video Tracking"

_remotesensing, doi:10.3390/rs15071735_

Round 1

Reviewer 1 Report

In this article, the authors propose a spectral-spatial transformer fusion method for hyperspectral video tracking, in which multiscale spectral-spatial features are extracted by two parallel branches and these features are fused by hyperspectral self-attention and hyperspectral cross-attention to capture the self-context feature interaction and the cross-context feature interaction. The proposed method employs CNN and Transformer for feature extraction and information exchange, respectively, which aims to solve the problem of global information utilization in hyperspectral video tracking. Some experimental results conducted on benchmark hyperspectral video tracking datasets have demonstrated the effectiveness of this method. The structure of this manuscript is relatively complete, and the conclusions are consistent with arguments. 

  And there are some problems to be solved. 1. Although this manuscript is readable, there are some obvious spelling errors in this manuscript, so the entire text should be checked carefully. 2. The proposed approach employs CNN and Transformer for feature extraction and information exchange, in the current manuscript, this method just combines existing CNN and Transformer models. The motivations and innovations should be further emphasized. 3. Hyperspectral video tracking is a very important research field, and the proposed method utilizes two types of neural networks to solve the problem of global information utilization. Some new tracking methods and datasets should be further compared and analyzed.   (such as, SiamHYPER: Learning a Hyperspectral Object Tracker From an RGB-Based Tracker;   Unsupervised Deep Hyperspectral Video Target Tracking and High Spectral-Spatial-Temporal Resolution (H³) Benchmark Dataset) 4. The model complexity is an important index to evaluate deep neural networks, and the model complexity of the proposed model should be further analyzed and compared with other tracking methods. 5. Some new hyperspectral video tracking approaches are not analyzed in this manuscript, more new references should be added. 6. It is recommended to analyze the depth and structure of the network, because different data sets correspond to different network structures.

Reviewer 2 Report

This paper proposed a spectral-spatial transformer fusion method for hyperspectral video tracking. The topic is a hot reserch point in remote sensing, and the spatial spectral transformer is also novel for target tracking. In general, there is no concern on the methdology and experiments. The manuscript can be accpeted in the current form.

Reviewer 3 Report

The authors have proposed a spectral-spatial transformer based feature Fusion tracker (SSTFT) for hyperspectral video tracking. The manuscript is complete, and the authors try to prove the progressiveness of the algorithm through experiments. However, there are some problems that need to be revised. The comments are as follows

1.      First, the computational complexity of the algorithm needs to be analyzed and compared with SOTA algorithm.

2.      I suggest reducing the contribution of the paper to three.

3.      Thirdly, some more methods regarding hyperpectral image should be investigated in your introduction, e.g., Semi-Supervised Locality Preserving Dense Graph Neural Network With ARMA Filters and Context-Aware Learning, Unsupervised Self-correlated Learning Smoothy Enhanced Locality Preserving Graph Convolution Embedding Clustering, Self-supervised Locality Preserving Low-pass Graph Convolutional Embedding, AF2GNN: Graph Convolution with Adaptive Filters and Aggregators, Multi-feature Fusion: Graph Neural Network and CNN Combining, MultiReceptive Field: An Adaptive Path Aggregation Graph Neural Framework.

4.      Fourthly, how about the adaptability of the algorithm to different number of training labels, especially small labels. Please compare with the SOAT methods.

5.      What is the adaptability of the algorithm proposed by the authors to image noise? Please use experiments to prove the progressiveness of the algorithm. That is to say, what is the classification performance of the algorithm when images are injected with different noises. In addition, when the classes are unbalanced, what is the classification effect of the algorithm.

6.       How the anchor box is generated in the algorithm needs further description and explanation. In addition, whether the authors have used different loss functions, I believe that improving the loss function can improve the performance of the algorithm.

Round 2

Reviewer 1 Report

I have no further comments, thanks for the revision of the authors.

Reviewer 3 Report

No more comments.